# Coupled reaction equilibria enable the light-driven formation of metal-functionalized molecular vanadium oxides

Stefan Repp [1,4], Moritz Remmers[2,4], Alexandra Stefanie Jessica Rein[1,4], Dieter Sorsche[1], Dandan Gao[2], Montaha Anjass[1,3], Mihail Mondeshki[2], Luca M. Carrella [2], Eva Rentschler [2] & Carsten Streb [1,2] ✉

The introduction of metal sites into molecular metal oxides, so-called poly-oxometalates, is key for tuning their structure and reactivity. The complex mechanisms which govern metal-functionalization of polyoxometalates are still poorly understood. Here, we report a coupled set of light-dependent and light-independent reaction equilibria controlling the mono- and di-metal-functionalization of a prototype molecular vanadium oxide cluster. Comprehensive mechanistic analyses show that coordination of a $Mg^{2+}$ ion to the species $\{(NMe_2H_2)_2[V^V_{12}O_{32}Cl]\}^{3-}$ results in formation of the mono-functionalized $\{(NMe_2H_2)[(MgCl)V^V_{12}O_{32}Cl]\}^{3-}$ with simultaneous release of a $NMe_2H_2^+$ placeholder cation. Irradiation of this species with visible light results in one-electron reduction of the vanadate, exchange of the second $NMe_2H_2^+$ with $Mg^{2+}$, and formation/crystallization of the di-metal-functionalized $[(MgCl)_2V^{IV}V^V_{11}O_{32}Cl]^{4-}$. Mechanistic studies show how stimuli such as light or competing cations affect the coupled equilibria. Transfer of this synthetic concept to other metal cations is also demonstrated, highlighting the versatility of the approach.

Molecular metal oxides, so-called polyoxometalates (POMs)[1,2], are an emerging class of inorganic materials. Their well-defined molecular structure and tuneable properties and reactivity have made POMs ideal analogues for the corresponding solid-state metal oxides[3–6]. This has led to applications in areas including energy conversion and storage[7,8], molecular magnetism[9,10], and catalysis[11,12]. Typically, POM reactivity can be tuned by functionalization with functional metal centres[13,14]. In molecular magnetism, the incorporation of lanthanides in POMs is a widely used principle[15], while energy conversion and storage schemes often use transition metal functionalized POMs[16]. In catalysis, the introduction of Lewis-acidic sites in POMs is a well-known concept[17].

Thus, robust and broad-scope synthetic concepts for the metal-functionalization of POMs are critical for the targeted development of new functional compounds. For polyoxotungstates (and, to a lesser degree, for polyoxomolybdates), controlled metal-functionalization is possible using so-called lacunary cluster derivatives where one or several of the original metal ions (W or Mo) have been hydrolytically removed from the cluster shell[14]. Selective binding of suitable metal ions at the resulting vacancy then leads to the targeted metal-functionalized species[14,18]. This approach has led to ground-breaking molecular components, e.g., for catalysis[11,16,19] molecular electronics[15,20,21] and medicine[22,23]. In contrast, in polyoxovanadate (POV) chemistry, controlled and predictable metal-functionalization approaches are still in their infancy[8,24–28], and the field is currently dominated by a combination of empirical knowledge and serendipity[8,24,25,29]. Thus, the development of controlled approaches for the predetermined and selective metal-functionalization of POMs is critical to enable knowledge-based materials design.

[1]Institute of Inorganic Chemistry I, Ulm University, Albert-Einstein-Allee 11, 89081 Ulm, Germany. [2]Department of Chemistry, Johannes Gutenberg University Mainz, Duesbergweg 10-14, 55128 Mainz, Germany. [3]Department of Chemistry, University of Sharjah, Sharjah-27272, Sharjah, United Arab Emirates. [4]These authors contributed equally: Stefan Repp, Moritz Remmers, Alexandra Stefanie Jessica Rein. ✉e-mail: carsten.streb@uni-mainz.de

Early studies by Streb and co-workers have used a placeholder strategy for the predictable metal-functionalization of POVs. This approach uses the dodecanuclear species $\{(NMe_2H_2)_2[V_{12}O_{32}Cl]\}^{3-}$, (= $(DMA)_2\{V_{12}\}$, DMA = dimethyl ammonium), where two vacant metal coordination sites are blocked by hydrogen-bonded DMA cations. In-situ exchange of these cations with a variety of metals is possible, leading to the mono- or (more rarely) di-metal-functionalized species ($\{MV_{12}\}$ and $\{M_2V_{12}\}$, respectively), and applications ranging from (light-driven) catalysis[30,31] to energy storage[32,33]. Note that for most of the di-metal-functionalized $\{V_{12}\}$ species were only accessible as mixed $V^{IV/V}$ oxidation state clusters[33–36]. Recently, a similar synthetic approach to metal-functionalized molybdates has been pioneered by Yamaguchi, Suzuki and co-workers. They used pyridine moieties to coordinatively stabilize and functionalize lacunary polyoxomolybdates which are otherwise difficult to access[18,37].

Most often, temperature, solvent, solution acidity and type of metal salt are the key synthetic parameters varied to facilitate metal-functionalization of POVs[24]. In contrast, the use of light, *i.e.* photons with energies in the visible spectral range has rarely been discussed as a systematic control parameter to trigger POM and POV functionalization. This is surprising, as the (visible) light photoactivity of POMs is well-documented, and the light-induced excitation of O→M ligand-to-metal charge-transfer transitions is an easy tool to selectively access reduced POM species[8,38–41]. The concept has been pioneered by Yamase and co-workers who demonstrated that previously unknown mixed-valent POMs and POVs can be accessed photochemically in the presence of suitable electron donors, e.g., organic amines[42–44]. POV chemistry is particularly sensitive to photoinduced reactions triggered by visible light, as POVs typically show higher visible light absorption compared with tungstates and molybdates[38]. This approach has recently been explored by Liu and co-workers who demonstrated the visible-light-assisted synthesis of mixed-valent POVs[45]. Here, we demonstrate how light-dependent, coupled reaction equilibria can be used to selectively target the partial reduction and metal-functionalization of $\{V_{12}\}$, leading to a di-magnesium functionalized species, $\{Mg_2V_{12}\}$, as an intriguing model compound for future studies, e.g., in electrochemical energy storage[8,33].

## Results

### Synthesis and characterization

The starting point was our study into the design of magnesium(II)-functionalized $\{V_{12}\}$ as molecular models for Mg ion batteries, where solid-state magnesium vanadates are under investigation as active electrode materials[46]. Initial experiments to functionalize $\{V_{12}\}$ with $Mg^{2+}$ were performed by reacting $Mg^{2+}$ with $(DMA)_2\{V_{12}\}$ in acetonitrile under the standard placeholder-functionalization conditions described above[13,34]. Despite the extensive variation of the reaction and isolation conditions, reproducible formation of Mg-functionalized $\{V_{12}\}$ was not possible. Systematic study of the key reaction parameters showed that visible light irradiation and oxygen-free conditions were required to access the target compound, $(nBu_4N)_4[(MgCl)_2V_{12}O_{32}Cl]$ (=$\{Mg_2V_{12}\}$, **1**). Diffusion crystallization using diethyl ether as diffusion solvent gave green single crystals of $\{Mg_2V_{12}\}$ in yields of 64% (based on $\{V_{12}\}$, see Methods and Supplementary Section 2). When the reaction was performed in the dark under otherwise identical conditions, only the starting material $\{V_{12}\}$ was recovered as yellow crystals. Crystallographic analysis by single-crystal X-ray diffraction shows that $\{Mg_2V_{12}\}$ crystallizes in the monoclinic space group $P2_1/c$ with cell axes $a = 24.3414(9)$ Å, $b = 16.7474(7)$ Å, $c = 24.4623(9)$ Å and cell angles $\beta = 94.6107(17)°$, $\alpha = \gamma = 90°$ (for crystallographic details see Methods and Supplementary Section 2.9). Note that this crystal lattice is virtually identical to the previously reported di-functionalized species $\{Mn_2V_{12}\}$ $(nBu_4N)_4[(MnCl)_2V_{12}O_{32}Cl]$[34]. For full characterization of $\{Mg_2V_{12}\}$ see Supplementary Section 2. The metal oxo framework of $\{Mg_2V_{12}\}$ is isostructural to the di-metal-functionalized $\{V_{12}\}$ species

reported earlier (i.e., $\{Mn_2V_{12}\}$[34], $\{Ca_2V_{12}\}$[33], $\{K_2V_{12}\}$[32], $\{Sr_2V_{12}\}$[35], $\{Ce_2V_{12}\}$[30]), the two square-pyramidal $Mg^{2+}$ ions reside in the metal binding sites on top and bottom of the cluster and feature a terminal chloride ligand (Fig. 1a, b).

UV-Vis-NIR spectroscopy of $\{Mg_2V_{12}\}$ confirms the mixed valent ($V^{IV/V}$) character of the species, as indicated by the characteristic, broad intervalence charge-transfer (IVCT) band between ~600–1200 nm[34]. Further support of the mixed-valent nature of $\{Mg_2V_{12}\}$ is given by continuous wave electron paramagnetic resonance (EPR) spectroscopy which unambiguously shows the presence of one $V^{IV}$ species with S = ½ (Supplementary Fig. 3), while a pure $V^V$ cluster would be EPR silent. Furthermore, EPR-based spin counting is in good agreement with one $V^{IV}$ centre per $\{Mg_2V_{12}\}$.

### Photochemical studies

Irradiation of a reaction mixture containing $\{V_{12}\}$ and $Mg^{2+}$ in MeCN with a broadband LED light source resulted in the emergence of the IVCT transitions characteristic for the formation of mixed-valence $VI^{IV/V}$ species (*vide supra*). Thus, the change of the UV-Vis-NIR signals over time can be used to monitor the rate of $\{Mg_2V_{12}\}$ formation, see Fig. 1c, d. This provides the ideal conditions to explore the fundamentals of the light-induced formation mechanism of metal-functionalized vanadates: to this end, we compared the photoreduction of $\{V_{12}\}$ in the presence and absence of $Mg^{2+}$ (Fig. 1d). Strikingly, $\{V_{12}\}$ reduction is only observed in the presence of $Mg^{2+}$, while in the absence of $Mg^{2+}$, no formation of $V^{IV}$ centres and no IVCT signal is detected. We hypothesized that this finding indicates that $Mg^{2+}$ interacts with $\{V_{12}\}$ in the reaction solution to give a photoactive reactive intermediate. Based on our understanding of the system, we suggested that this intermediate could be the mono-functionalized species $\{MgV_{12}\}$ (= $\{(DMA)[(MgCl)V^V_{12}O_{32}Cl]\}^{3-}$). The formation of an intermediate species is indicated by UV-Vis-NIR spectroscopy, which shows distinct spectral changes in the region between 300 nm to 500 nm upon addition of $Mg^{2+}$ to the $\{V_{12}\}$ reaction solution (Fig. 2a). Also, $^{51}V$ NMR spectroscopy shows that immediately after addition of $Mg^{2+}$ to an acetonitrile solution of $\{V_{12}\}$, a four-signal spectrum is observed which is characteristic for the mono-metal-functionalized $\{MV_{12}\}$ species (Fig. 2b).

Note that a virtually identical four-line signal pattern has been reported for the mono-$Zn^{2+}$-functionalized species $\{ZnV_{12}\}$ (= $\{(DMA)[(ZnCl)V_{12}O_{32}Cl]\}^{3-}$)[13]. In addition, $^{51}V$ NMR total correlation spectroscopy (TOCSY) of the reaction solution showed, that the four $^{51}V$ NMR signals assigned to $\{MgV_{12}\}$ belong to one molecular species (Fig. 2c).

### Mechanistic analyses

Further evidence for the formation of $\{MgV_{12}\}$ is revealed by characteristic changes in the respective $^1H$ and $^1H$ DOSY spectra (Supplementary Section 3.1). Upon addition of $Mg^{2+}$ to a $\{V_{12}\}$ solution in acetonitrile, DMA cations are released from their original, $\{V_{12}\}$-bound positions into solution, resulting in a dynamic equilibrium between cluster-bound and "free" DMA cations. This results in a characteristic low-field shift of the N-H proton resonances from $\delta$ ~ 6.3 ppm to $\delta$ ~ 7.0 ppm (Supplementary Fig. 12 and Fig. 13). $^1H$ DOSY NMR spectra were collected to further study the release of DMA cations upon $Mg^{2+}$ addition to $\{V_{12}\}$ solutions in acetonitrile. Specifically, we studied the characteristic changes of the respective diffusion coefficients $D$ based on analysis of the DMA–$^1H$ resonances (methyl groups, $\delta$ *ca.* 2.6 ppm; ammonium groups, $\delta$ *ca.* 6.3 – 8.6 ppm, (Supplementary Fig. 14)[47]. These analyses show the expected trend, *i.e.*, the diffusion coefficients decrease with increasing size of the species studied in the order "free" DMA <$\{MgV_{12}\}$ <$\{V_{12}\}$.

Further, high-resolution electrospray ionization mass spectrometry (HR ESI MS) allowed us to identify a series of signals corresponding to the mono-Mg-functionalized species, e.g., $[HMgV_{12}O_{32}Cl]^{2-}$ (observed: 591.590 m/z, calculated: 591.852 m/z, see Fig. 2d, and Supplementary Section 2.8 for further peak assignments).

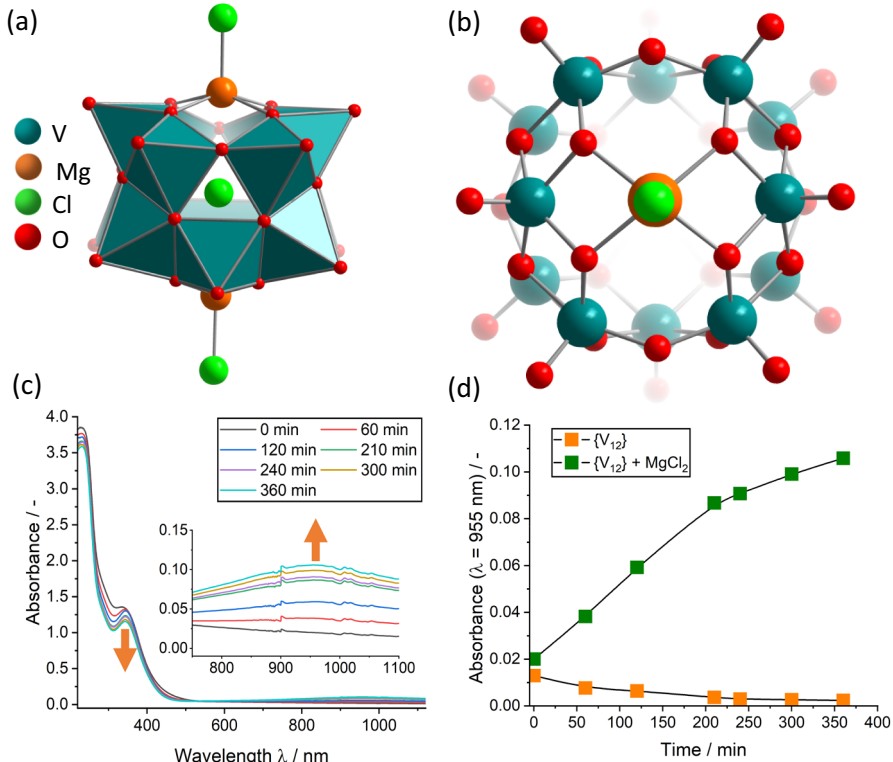

**Fig. 1 | Structural and spectroscopic information on the formation of {Mg₂V₁₂}.**
**a** side view of {Mg₂V₁₂}; (**b**) top view of the Mg binding site in {Mg₂V₁₂}; (**c**) time-lapse UV-Vis-NIR spectroscopy of the {Mg₂V₁₂} reaction mixture containing {V₁₂} and MgCl₂ in acetonitrile. **d** time-dependent reduction of {V₁₂} in the presence

(green squares) and absence (orange squares) of Mg²⁺. Conditions: irradiation with a broadband high-power LED light source ($P_{optical}$ ~ 5 W), [{V₁₂}] = 0.05 mM, [Mg²⁺] = 0.21 mM.

Note that the characteristic four-line $^{51}$V NMR signal pattern assigned to {MgV₁₂} was observed even under the dilute concentration conditions of the HR ESI MS experiments ([Mg²⁺] = 0.21 mM, [{V₁₂}] = 0.05 mM, see Supplementary Fig. 15).

Next, we explored the Mg²⁺ functionalization further by performing a $^{51}$V NMR spectroscopy titration, where increasing amounts of Mg²⁺ were added to {V₁₂} solutions in acetonitrile. To assess the formation of {MgV₁₂}, the characteristic $^{51}$V NMR signals were integrated, and the integral areas were plotted as a function of the Mg²⁺ equivalents added. As shown in Fig. 3, integration of the three non-overlapping signals unambiguously indicates, that changes of the integrated area are only observed up to 1.0 equivalents Mg²⁺. Higher equivalents do not change the spectra observed. This strongly suggest the presence of a 1:1 molar species, which is in line with the formation of {MgV₁₂}. These observations are supported by an identical $^{1}$H NMR titration study which shows that upon Mg²⁺ binding to {V₁₂}, release of DMA cations (indicated by characteristic shifts of the DMA proton signals) is observed, see Supplementary Fig. 13.

The different photoactivities of {V₁₂} and {MgV₁₂} were probed experimentally by wavelength-selective irradiation: when the standard {V₁₂}/Mg²⁺ reaction mixture in acetonitrile was irradiated with a monochromatic 405 nm LED light source, the characteristic {MgV₁₂} reduction and formation of the characteristic IVCT band was observed. In contrast, when the same experiment was performed for a pure {V₁₂} solution (in acetonitrile, without added Mg²⁺), no vanadate reduction was observed. Also, irradiation of the standard {V₁₂}/Mg²⁺ reaction mixture using a monochromatic 470 nm LED light source also did not lead to reduction of the vanadate cluster, see Supplementary Fig. 18.

Further insights into the electronic structure of {MgV₁₂} and {Mg₂V₁₂} were obtained by theoretical computations using density functional theory (DFT) using the B3LYP functional[48,49] combined with

the def2-SVP basis set[50]. Analysis of the HOMO-LUMO levels of {MgV₁₂} and analysis of the calculated UV-Vis-NIR spectrum show an intense ligand-to-metal-charge-transfer (LMCT) transition at the UV-to-Vis border, which we attribute to the experimentally observed Vis photoactivity of {MgV₁₂}. For {Mg₂V₁₂}, similar LMCT transitions are observed, and in addition, the broad characteristic IVCT transition in the Vis-NIR range are reproduced by the calculations. For details, see Supplementary Section 4.

Based on Le Chatelier's principle, we hypothesized that addition of DMA to the reaction solution should shift the equilibrium to the reagent side (see Fig. 4a), thus preventing the formation of the photoactive {MgV₁₂}. This behaviour is indeed observed: when an excess of DMACl is added to the standard {V₁₂} reaction solution and the sample is irradiated, no reduction is observed by UV-Vis-NIR spectroscopy (Supplementary Fig. 19). This suggests that the reactive intermediate which enables photoreduction is not present under these conditions and lends further support to {MgV₁₂} being the photoactive intermediate. Also, when the {V₁₂} photoreduction is performed in the presence of Mg²⁺ and air, virtually no vanadate reduction is observed (as indicated by the absence of IVCT bands in the UV-Vis-NIR spectrum, see Supplementary Fig. 20). This provides further support that a light-induced electron transfer to the photoexcited {MgV₁₂} is a key process in the formation of {Mg₂V₁₂} and suggests that interference between the photoexcited {MgV₁₂} and O₂ (e.g., by triplet quenching[51]) could prevent formation of the reduced vanadate species[52].

In sum, these data suggest that reaction of {V₁₂} and Mg²⁺ results in formation of the mono-functionalized {MgV₁₂} as photoactive, reactive intermediate, which can then be converted to the di-functionalized {Mg₂V₁₂} upon irradiation with visible light. To gain insights into the sacrificial electron donor, we used cyclic voltammetry to compare the redox potentials of the possible donors (DMA, $n$Bu₄N⁺, MeCN). Based

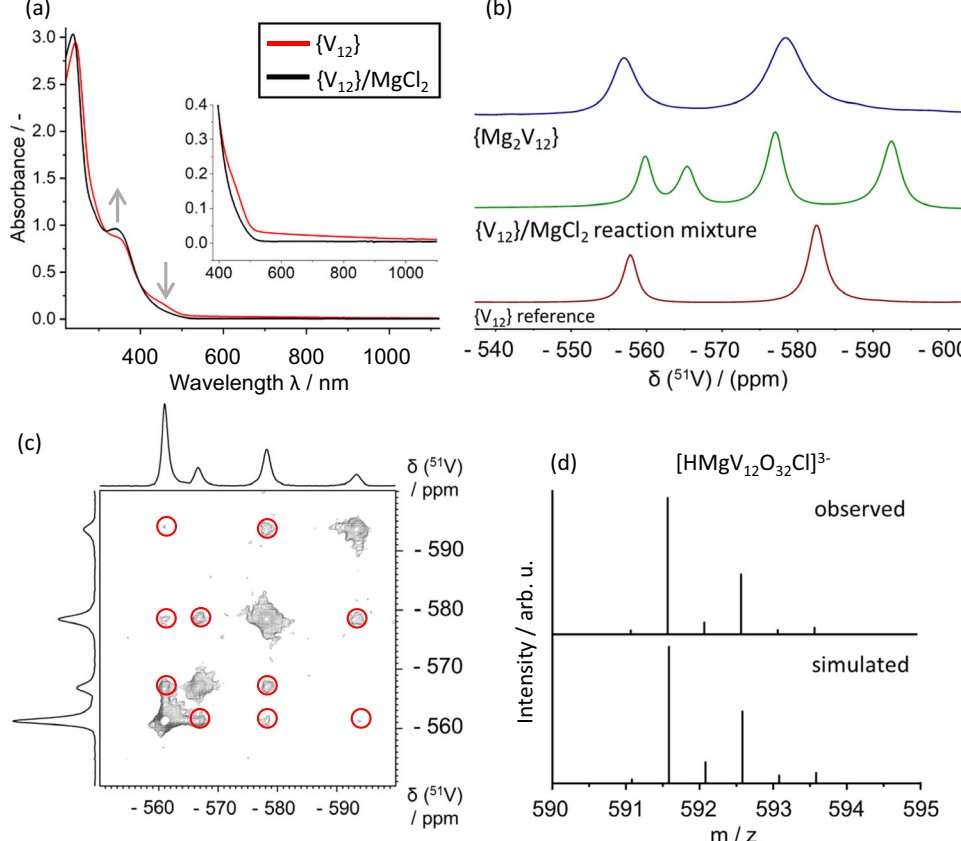

**Fig. 2 | Experimental verification for the in-situ formation of the photoactive {MgV$_{12}$} intermediate under reaction conditions. a** UV-Vis-NIR spectral changes observed upon reaction of {V$_{12}$} (0.05 mM) with Mg$^{2+}$ (0.21 mM) in acetonitrile, resulting in the formation of the visible-light photoactive {MgV$_{12}$}. **b** $^{51}$V NMR spectroscopic observation of the characteristic four-line pattern of mono-functionalized {MgV$_{12}$} formed by reaction of {V$_{12}$} (10 mM) with Mg$^{2+}$ (42 mM) in acetonitrile. $^{51}$V NMR spectra of {V$_{12}$} (in acetonitrile) and {Mg$_2$V$_{12}$} (in dimethyl sulfoxide) are shown for comparison. **c** $^{51}$V TOCSY NMR spectrum indicating that all four signals belong to one $^{51}$V spin system i.e., one {MgV$_{12}$} cluster. Non-diagonal signals are marked with red circles. Conditions: [{V$_{12}$}] = 5.0 mM, [Mg$^{2+}$] = 21.1 mM, solvent: acetonitrile. **d** Negative ion-mode high-resolution ESI mass spectrum showing the observed and simulated isotopic pattern for [HMgV$_{12}$O$_{32}$Cl]$^{2-}$ (= H{MgV$_{12}$}), [{V$_{12}$}] = 0.05 mM, [Mg$^{2+}$] = 0.21 mM, solvent: acetonitrile.

on this data, DMA is the most likely sacrificial electron donor, as it is significantly easier to oxidize ($E_{ox}$ = 0.7 V vs Fc$^+$/Fc) compared with the other possible electron donors, i.e., $n$Bu$_4$N$^+$ and MeCN ($E_{ox}$ > 1.6 V vs Fc$^+$/Fc), see Supplementary Fig. 21. Based on these considerations, the following coupled reaction equilibria are proposed, see Fig. 4.

Finally, to probe whether the observed light-induced reactivity is unique to Mg$^{2+}$ or can be extended to other metal functionalizations, we performed the {V$_{12}$} metal functionalization experiments using Ca$^{2+}$ instead of Mg$^{2+}$ (experimental details see Supplementary Section 3.4). Reaction of CaCl$_2$ x 2H$_2$O (21.1 mM) with {V$_{12}$} (5.0 mM) in acetonitrile led to the observation of the characteristic four-line $^{51}$V NMR signal pattern assigned to {CaV$_{12}$} (Supplementary Fig. 22). Visible light-irradiation of the {V$_{12}$} / Ca$^{2+}$ reaction solution resulted in the formation of the characteristic IVCT bands between ~600 and 1200 nm which is indicative of the formation of the reduced, di-metal-substituted vanadate species (see Supplementary Fig. 23). These findings suggest that the light-induced metal functionalization reported here is not unique to Mg$^{2+}$ and can be transferred to other metal cation species, and possibly also to other vanadate cluster architectures.

## Discussion

We report the first example of a light-dependent, coupled set of solution-phase equilibria, enabling the controlled metal-functionalization of molecular vanadium oxides. Light-independent reaction of {V$_{12}$} with Mg$^{2+}$ results in a dynamic pre-equilibrium, where one DMA placeholder cation on {V$_{12}$} is replaced with one Mg$^{2+}$ ion, resulting in formation of the Vis-photoactive intermediate {MgV$_{12}$}. The formation of a 1:1 species was verified by $^{51}$V NMR / $^1$H NMR spectroscopy as well as HR ESI MS studies. Competitive binding studies using DMA and Mg$^{2+}$ show, that this pre-equilibrium is sensitive to the Mg$^{2+}$ / DMA molar ratio, essentially allowing an on/off switching of the metal functionalization.

Visible light-irradiation of {MgV$_{12}$} solutions results in the one-electron photoreduction of the cluster, release of the second DMA placeholder cation, and binding of a second Mg$^{2+}$ metal centre, yielding {Mg$_2$V$_{12}$}. The increased visible-light photoactivity is in line with recent literature reports which show that metal-incorporation in POMs leads to a lowering of the HOMO-LUMO gap and thus, increased photoactivity[53]. The photoredox processes at {MgV$_{12}$} only occur in the absence of water and oxygen, indicating possible interference of these species with the light-induced electron transfer to the cluster. Electrochemical studies suggest that DMA is the most likely electron donor based on analysis of the redox potentials of the reagents used. Mechanistic experimental and theoretical studies show the light-dependent nature of the assembly process and emphasize how supramolecular reaction control can be used to trigger or inhibit photoactivity. Finally, initial experiments show that a similar route can be followed to enable {V$_{12}$} functionalization with Ca$^{2+}$ using light-induced cluster assembly. Thus, the principles outlined in this report open new paths for designing multi-stimuli-responsive molecular materials.

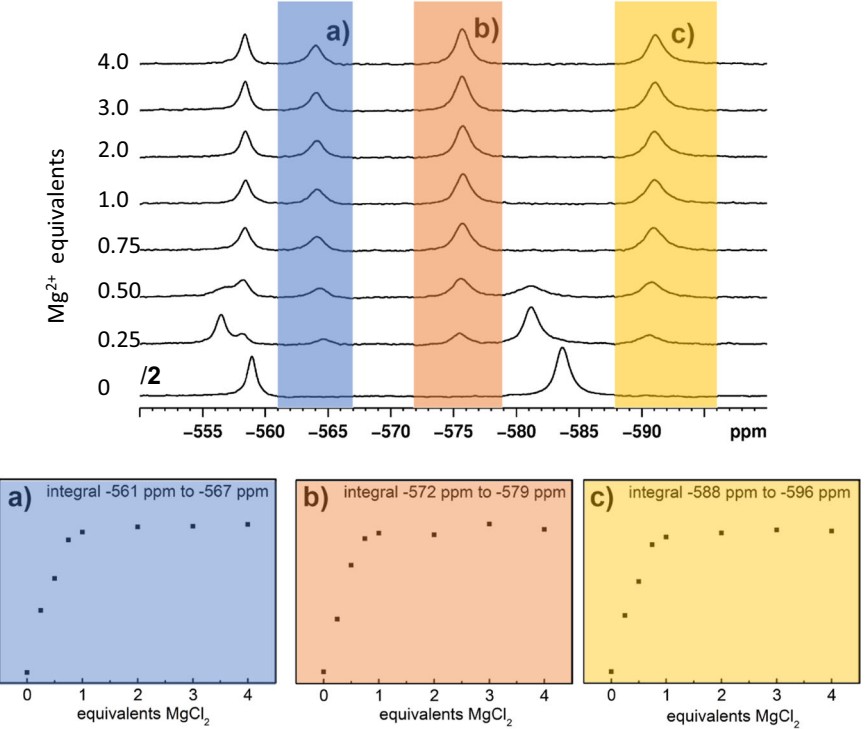

**Fig. 3 | In-situ $^{51}$V NMR spectroscopic titration to assess the {MgV$_{12}$} formation.**
Top: stacked $^{51}$V NMR spectra of acetonitrile solutions containing {V$_{12}$} and varying
Mg$^{2+}$ molar equivalents (between 0 eq. to 4 eq. relative to {V$_{12}$}). Bottom: the area
integrals of the three characteristic {MgV$_{12}$} signals marked (**a**–**c**) are shown. In each

instance, integral changes are only observed up to 1.0 Mg$^{2+}$ equivalents, indicating
that a 1:1 species, i.e., {MgV$_{12}$} is formed. Conditions: [{V$_{12}$}] = 5.0 mM,
[Mg$^{2+}$] = 0–21.1 mM, solvent = acetonitrile.

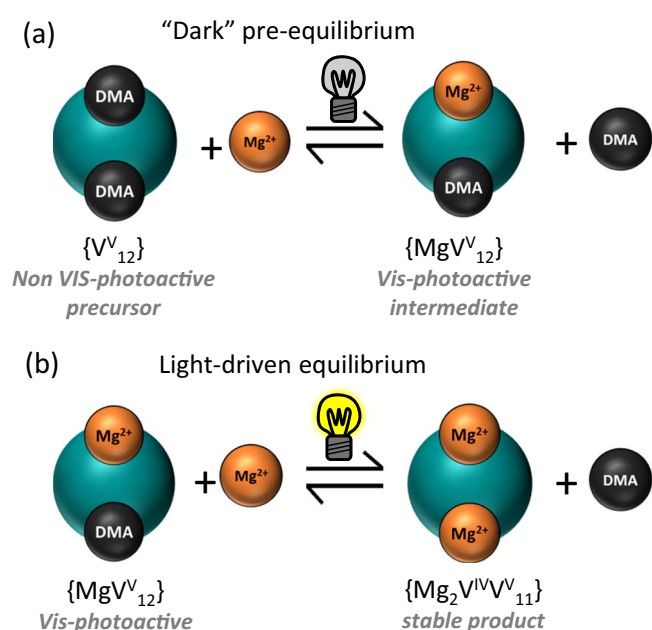

**Fig. 4 | Proposed coupled solution-phase equilibria during {Mg$_2$V$_{12}$} formation.**
**a** The light-independent pre-equilibrium forming the photoactive intermediate
{MgV$_{12}$} and (**b**) light-driven formation of the di-Mg-functionalized {Mg$_2$V$_{12}$}.

## Methods

### Synthesis of 1

($n$Bu$_4$N)$_4$[(MgCl)$_2$V$^{IV}$V$^V_{11}$O$_{32}$Cl] x CH$_3$CN (= ($n$Bu$_4$N)$_4${Mg$_2$V$_{12}$})

The synthesis of **1** was performed in a glovebox under argon
atmosphere: in a 25 mL round-bottom flask 0.200 g (0.100 mmol)

($n$Bu$_4$N)$_3$(NMe$_2$H$_2$)$_2$[V$_{12}$O$_{32}$Cl)] x CH$_3$CN, 0.056 g (0.200 mmol)
$n$Bu$_4$NCl and 0.0400 g (0.420 mmol) anhydrous MgCl$_2$ were dissolved
in 12 mL water-free, deaerated acetonitrile and stirred at room tem-
perature. After four hours of stirring, the yellow solution was filtered
through a glass Pasteur pipette filled with glass wool. Diffusion crys-
tallisation with diethyl ether was setup and the samples were exposed
to light via the glovebox fluorescent lamps or a broadband LED light
source (see Supplementary Fig. 1 for emission spectrum of the light
source). After two days, dark green crystals of **1** were obtained, filtered,
washed twice with acetone and diethyl ether and dried vacuum. Yield:
0.146 g (0.0634 mmol, 63.8% based on V). Characteristic IR bands (in
cm$^{-1}$): 3336; 2961 (C-H stretching, alkane); 2934 (C-H stretching,
alkane); 2874 (C-H stretching, alkane); 1659; 1641; 1613; 1482; 1461;
1381;1422; 1345; 1278; 1251; 1163; 1151; 1107; 1067; 995 (symmetric V = O,
terminals Oxygen); 890; 875 (anti-symmetric stretching VO unit); 817;
752 (V-O-V, symmetric); 661 (V$_3$-O$_{\mu3}$, asymmetric); 593; 412. UV-Vis-NIR
spectroscopic          maxima          ([{Mg$_2$V$_{12}$}  =   56 µM   in   DMF):
$\varepsilon_{338}$ = 11,875 M$^{-1}$ cm$^{-1}$; $\varepsilon_{991}$ = 1,400 M$^{-1}$ cm$^{-1}$. $^1$H NMR (400 MHz, DMSO-
d$_6$): δ (ppm) = 3.44 (s, H$_2$O); 3.18 (m, 2 H, $n$Bu$_4$N$^+$); 2.45 (s, unassigned);
2.07 (s, acetonitrile); 1.56 (m, 2 H, $n$Bu$_4$N$^+$); 1.30 (s, 2 H, $n$Bu$_4$N$^+$); 0.92 (t,
3 H, $n$Bu$_4$N$^+$). $^{51}$V NMR (105 MHz, DMSO-d$_6$): δ (ppm) = − 557.0 (s, 4 V);
−578.3 (s, 8 V).

### Crystallographic data for 1

($n$Bu$_4$N)$_4$[Mg$_2$Cl$_3$V$_{12}$O$_{32}$] x CH$_3$CN, (M = 2289.15 g/mol): monoclinic,
space group *P*2/c, *a* = 24.3414(9) Å, *b* = 16.7474(7) Å, *c* = 24.4623(9) Å,
β = 94.6107(17), V = 9939.9(7) Å$^3$, Z = 4, *T* = 150 K, ρ$_{calc}$ = 1.530 g/cm$^3$,
μ(MoKα) = 1.238 mm$^{-1}$, 332994 reflections measured, 22031 unique
(R$_{int}$ = 0.0771, R$_{sigma}$ = 0.0266), R$_1$ = 0.0518 (I > = 2σ(I)), wR$_2$ = 0.1449
(all data). CCDC 2240239 contains the supplementary crystallographic
data for this paper. These data can be obtained free of charge from The
Cambridge Crystallographic Data Centre via www.ccdc.cam.ac.uk/
structures.

## Solution-phase NMR spectroscopic analyses

Mechanistic reactivity studies using $^1$H and $^{51}$V NMR spectroscopy were carried by dissolving the compound under study in the respective solvent. The reactions were performed under the given conditions as stated above and in the Supplementary Information. All solutions were prepared in an argon-filled glovebox unless stated otherwise.

## Data availability

The datasets generated during and/or analysed during the current study are available in the zenodo.org repository and can be retrieved using the following link: https://doi.org/10.5281/zenodo.8316822. The crystallographic data reported for **1** (CCDC no 2240239) can be obtained free of charge from The Cambridge Crystallographic Data Centre via www.ccdc.cam.ac.uk/structures.

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

## Acknowledgements

Financial support by the Deutsche Forschungsgemeinschaft DFG, project no 390874152 (S.R., C.S.), 389183496 (C.S.), 404530119 (C.S.), 510966757 (D.G.) and the ERC Consolidator Grant "SupraVox", rant no: 101002212 (C.S.) is gratefully acknowledged. C.S. gratefully acknowledges the Gutenberg Research College Mainz and the Top-Level Research Area SusInnoScience for financial support. M.Sc. D. Kowalcyk and Prof. D. Ziegenbalg (Ulm University) are acknowledged for characterization of the emission features of the LED light source. Prof. Sven Rau (Ulm), M.Sc. Ludwig Schwiedrzik (Vienna University) and M.Sc. David Hernández-Castillo are gratefully acknowledged for constructive comments and discussions.

## Author contributions

S.R., D.S. and C.S. conceptualized the study. S.R., A.S.J.R., D.S. and M.R. performed syntheses and characterization. S.R. performed synthesis, crystallography, spectroscopy, and data analysis. M.M. performed and analyzed NMR spectroscopy. D.G. performed electrochemical experiments. M.A. performed computational analyses. L.M.C. and E.R. performed EPR spectroscopic analyses. All authors co-wrote the manuscript.

## Funding

## Competing interests
The authors declare no competing interests.
