## [Peer Review File · Nature Communications]

Coupled reaction equilibria enable the light-driven formation of metal-functionalized molecular vanadium oxidesReviewers' Comments:

Reviewer #1:

Remarks to the Author:

In this short report Streb and colleagues describe the results of the experimental observation of a photoinduced post-functionalization of a tubular V12-nuclearity polyoxovanadate compound by Mg²⁺ ions. The authors show that irradiation of a mono-functionalized MgV12 derivative with visible light gives rise to the formation of a di-functionalized Mg₂V12 compound. The manuscript is rather short but the methodology is sound. I found this study very interesting (also, in terms of far-reaching implication for bio-inspired ionic lock reactions) and well-structured. Overall, I can recommend this study for publication after a major revision.

1) I suggest to include the following relevant studies on metal-functionalization of tubular V12 structures into the reference list:

Werner et al., *Inorg. Chem.* 2023, 62, 9, 3761-3775

Pütt et al., *Chem. Commun.* 2019, 55, 13554-13557

2) Amine-containing species are capable of reducing polyoxovanadates in solution. Can the authors comment on the role of N-containing species as well as counteranions in the irradiation process of the title compounds?

3) I would appreciate if the authors consider elucidating the mechanism of light activated structural changes in solution by providing corresponding NMR measurements combined with MD simulations. Does this light-induced Mg-post-functionalization also take place in the solid state?

Reviewer #2:

Remarks to the Author:

The paper by Repp et al reports on the light-dependent and light-independent reaction equilibria that can be used to control the formation of mono- and di-metal-functionalization of a molecular vanadium oxide cluster. The authors performed comprehensive mechanistic analyses to prove the existence of such equilibria, and have successfully isolated and characterized the end product, which is a di-metal-functionalized species [(MgCl)₂V12O32Cl]₃. The paper is very well written, and the concept of using light to control the selective metal-functionalization of POMs is certainly highly interesting and novel, however some additional effort might be needed to confirm this novel hypothesis, as highlighted below.

1. The proposal that equilibria between V12 and Mg lead to the formation of mono-functionalized MgV12 species, which is photoactive, is very interesting but also very intriguing. Although V-51 NMR and ESI-MS indeed suggest that MgV12 species might be formed, closer inspection of these spectra raise some questions. A four-signal spectrum in Figure 2a is indeed indication that some asymmetrical species is formed, but how can we be sure that it is indeed mono-substituted MgV12 species? Some reference to the literature would be helpful here to support the claims given. Inspection of ESI-MS spectra shown in Figure S13 indicates the presence of many other vanadate species in solution, and compared to them, mono-substituted MgV12 is present in relatively small amount. Are these vanadate species also seen in the V-51 NMR? Can the four-signal spectrum be assigned to other species that are seen in ESI-MS, or even to a mixture of two or more vanadate species?

2. In Figure 2, the concentrations of each component (V12 and Mg salt) should be given for spectra a and c. I assume that concentrations used for 51-V NMR are much higher than those used for UV-Vis? Could this influence solution equilibria? In other words, while 51-V NMR suggests the presence of MgV12 (Fig 2a) (see point above), the existence of such species is much less obvious in UV-Vis spectrum shown in Fig 2c. I could imagine that in diluted solutions the binding between V12 and

Mg(II) would be less favorable? Could changes observed in the UV-Vis spectra be related to the presence of other vanadate species detected in ESI-MS?

3. Very little information is provided on the time scale of MgV12 species formation and its conversion to the final Mg2V12 form. The fact that both V-51 and ESI-MS could be recorded implies that this intermediate is sufficiently stable. How quickly does it convert to Mg2V12? According to Figure 3, it should be simple exposure to the light that should convert these species to Mg2V12? I believe such experiment was attempted in Figure S5, but it seems that irradiation for 3 or 24 h under oxygen-free condition does not produce Mg2V12. How can this be explained? And how does this relate to the proposed mechanism in Figure 3?

4. Figure S6 shows changes in the ^1H NMR spectrum of DMA upon addition of MgCl to V12. Why is there such a large difference seen in the DMA peak position after irradiation of 3h or 24h, when no change is to be seen in the ^{51}V NMR spectra shown in Figure S5 for such solutions?

5 Related to the point above, studying this conversion is crucial to confirm the second step (i.e. the light-driven equilibria) shown in Figure 3, as I am bit missing unambiguous evidence that step b in Figure 3 really occurs as shown here. The UV/Vis spectra shown in Fig1 indicate that irradiation of V12/MgCl2 mixture indeed results in formation of the reduced Mg2V12, but what is the evidence that MgV12 is indeed seen in this spectrum as the starting material for this conversion? (btw, there seem to be something wrong with y-axis in Figures 1c and 1d).

6 The idea of using excess of DMA to probe the equilibrium shown in Figure 3 is a clever one, but based on Le Chatelier's principle the concentration of Mg(II) should also have profound influence on these equilibria shown in Figure 3. Has the effect of Mg(II) concentration on the equilibria in Fig 3 been investigated?

7. The authors mention in the abstract that the addition of other cations which compete with Mg(II) can effectively inhibit the formation of metal-functionalized clusters. However, besides the experiments with DMA mentioned above, I can't seem to find any competition experiments with other metal cations. I think it would be highly interesting to explore this, also in order to prove whether it is really something "special" about Mg(II) that requires the light driven equilibria to form di-substituted V12 species, or is this phenomenon pertinent to other metal cations?

8. In few places where the concentration of Mg(II) was mentioned (for example in Figure 1 captions) it seems that Mg(II) was used in more than four-fold excess for the formation of Mg2V12, although two fold excess would have been sufficient. Why was the excess of Mg necessary? What was the ratio of Mg(II)/V12 in the experiments where the formation of the mono-substituted MgV12 was probed?

9. The paper is missing in-depth discussion of this novel phenomenon. The section marked as "Discussion" is actually only a brief conclusion. A more thorough discussion should be given, as it is very intriguing that binding of Mg(II), which is a redox inactive metal, can result in formation of photoactive intermediate. What is a possible explanation/rationale for this? How does the formation of Mg2V12 differ from the formation of other di-metal substituted V12 species mentioned in the introduction? Is this reaction the only one that requires presence of light and the absence of oxygen? Is the light driven equilibrium an oddity that is specific for this Mg(II)/V12 system or maybe a more general phenomenon that can be explored in the future?

10. It is well known that metal substitution of POMs lowers HOMO/LUMO energy resulting in visible light photoactive POM species (ACS Catal 2018, 8 (11), 10809–10825). In that respect, theoretical calculations would have been useful to understand the nature of the novel phenomenon reported here.

Reviewer #3:

Remarks to the Author:

This is an important and creative study that reports a new phenomenon in the synthesis of molecular metal oxide clusters. The photoactivation of the V12 molecular building block allows access to an otherwise inaccessible di-functionalised compound. The work has implications for synthetic chemists working in molecular metal oxide chemistry, and potentially even for those working in the preparation of extended metal oxides.

The conclusions presented in this study are consistent with the data presented and the methodology appears sound. The experimental details are coherent and complete, and I am confident that the work could be recreated. I am happy to recommend publication of this paper once the following points have been addressed:

The charge of the {V12} precursor compound $(\text{NMe}_2\text{H}_2)_2[\text{V}_{12}\text{O}_{32}\text{Cl}]^{3-}$ introduced on page 2 is difficult to understand. I assume the 3- applies to the whole assembly, but this should then be clarified (perhaps with curly brackets to emphasise that this is a single anionic complex). Considering this, it might be worth including the V oxidation states in the formulae throughout the manuscript – this would help the reader to follow the redox chemistry at play.

The use of a visible light source feels like a bit of a blunt instrument here. Can the authors please add a comment on the choice of wavelengths used, and any wavelength-selective observations?

Page 5 – where the authors state the redox potential of $\text{O}_2/\text{H}_2\text{O}$ as +0.6 V – This E^0 value (note, this is the formal potential, not the $E_{1/2}$ value as quoted in the manuscript) is based on a system where O_2 is at 1 atm and H^+ and H_2O are at unit activity in the organic solvent (actual concentrations used were 1 M H^+ and 1 M H_2O). The authors data appears to have been collected from 1M $(\text{nBu}_4)\text{PF}_6$ in DMF. The value comparison quoted, therefore is meaningless and should be replaced with system specific data. The absence of protons in the sample voltammetry conditions means that it's likely that oxygen reduction occurs via the single electron radical pathway. It's still likely that this will reoxidize the reduced V12 compound, but real electrochemical data here would be a welcome addition.

Have the authors considered what is providing the electron in the reduction of the cluster? This is an important point and should be discussed in the manuscript. Does changing the solvent system impact the reaction process? I would expect DMF to be a relatively effective electron donor and may therefore improve efficiency/rate of the reaction.

REVIEWER COMMENTS (Replies given in blue font)

Reviewer #1 (Remarks to the Author):

In this short report Streb and colleagues describe the results of the experimental observation of a photoinduced post-functionalization of a tubular V₁₂-nuclearity polyoxovanadate compound by Mg²⁺ ions. The authors show that irradiation of a mono-functionalized MgV₁₂ derivative with visible light gives rise to the formation of a di-functionalized Mg₂V₁₂ compound. The manuscript is rather short but the methodology is sound. I found this study very interesting (also, in terms of far-reaching implication for bio-inspired ionic lock reactions) and well-structured. Overall, I can recommend this study for publication after a major revision.

Q1: I suggest to include the following relevant studies on metal-functionalization of tubular V₁₂ structures into the reference list:

Werner et al., Inorg. Chem. 2023, 62, 9, 3761-3775

Pütt et al., Chem. Commun. 2019, 55, 13554-13557

Reply1: we have added these studies at appropriate sections in the main manuscript.

Q2: Amine-containing species are capable of reducing polyoxovanadates in solution. Can the authors comment on the role of N-containing species as well as counter-cations in the irradiation process of the title compounds?

Reply2: we have analyzed the possible electron donors present in the reaction solution (dimethyl ammonium (DMA), *n*Bu₄N⁺ and acetonitrile). In sum, DMA has the least positive redox potential ($E_{ox} = +0.7$ V vs Fc⁺/Fc under the standard reaction conditions employed). This redox potential is constant and does not depend on whether DMA is bound to {V₁₂} or "free" in solution. The counter-cations (*n*Bu₄N⁺) and solvent (acetonitrile) used in the studies have significantly more positive redox potentials ($E_{ox} > 1.6$ V vs Fc⁺/Fc) and are oxidatively significantly more stable than DMA. Based on these data we suggest that DMA could be the sacrificial electron donor under the given conditions. This is now described in the manuscript and SI (Supplementary Section 3.3).

Q3: I would appreciate if the authors consider elucidating the mechanism of light activated structural changes in solution by providing corresponding NMR measurements combined with MD simulations. Does this light-induced Mg-post-functionalization also take place in the solid state?

Reply3: the current manuscript proposes an initial mechanism for the observed metal-functionalization based on experimental data (NMR, ESI MS, UV-Vis, XRD). In the revision, we also now provide initial DFT-level computations on the electronic structure changes of the system during the light-induced conversion. However, a molecular dynamics (MD) simulation of these light-induced processes is very difficult, as

(a), it would require the development of specific reactive force-fields (ReaxFF method) to describe structural changes of the {V₁₂} system, as standard force fields cannot model the making or breaking of bonds. These ReaxFF models currently do not exist for polyoxometalates; in addition, even classical force-fields do not exist for this specific class of polyoxovanadates. For POMs in general, development of such force fields requires > 6 months development time by expert computational groups.

(b) MD is generally rather non-suitable to model photochemical processes, as excited states would have to be considered, which is not feasible in MD and very difficult, even at DFT level.

With respect to possible solid-state conversions, we have not observed any light-induced reactivity for the solid-state samples of {V₁₂}.

Reviewer #2 (Remarks to the Author):

The paper by Repp et al reports on the light-dependent and light-independent reaction equilibria that can be used to control the formation of mono- and di-metal-functionalization of a molecular vanadium oxide cluster. The authors performed comprehensive mechanistic analyses to prove the existence of such equilibria, and have successfully isolated and characterized the end product, which is a di-metal-functionalized species $[(\text{MgCl})_2\text{V}_{12}\text{O}_{32}\text{Cl}]^{3-}$. The paper is very well written, and the concept of using light to control the selective metal-functionalization of POMs is certainly highly interesting and novel, however some additional effort might be needed to confirm this novel hypothesis, as highlighted below.

Q4: The proposal that equilibria between V12 and Mg lead to the formation of mono-functionalized MgV12 species, which is photoactive, is very interesting but also very intriguing. Although V-51 NMR and ESI-MS indeed suggest that MgV12 species might be formed, closer inspection of these spectra raise some questions. A four-signal spectrum in Figure 2a is indeed indication that some asymmetrical species is formed, but how can we be sure that it is indeed mono-substituted MgV12 species? Some reference to the literature would be helpful here to support the claims given. Inspection of ESI-MS spectra shown in Figure S13 indicates the presence of many other vanadate species in solution, and compared to them, mono-substituted MgV12 is present in relatively small amount. Are these vanadate species also seen in the V-51 NMR? Can the four-signal spectrum be assigned to other species that are seen in ESI-MS, or even to a mixture of two or more vanadate species?

Reply4: the assignment of the ^{51}V NMR data and the ESI MS data to the **{MgV12}** species is now clarified in the manuscript and SI. We have performed additional ^{51}V TOCSY NMR measurements (TOCSY - total correlation spectroscopy), which verify that all four signals observed in the ^{51}V NMR spectra of **{MgV12}** belong to one molecular species, which is in line with the high-resolution ESI MS Data presented. In addition, previous works have reported nearly identical four-line ^{51}V NMR spectra for other, isolated mono-metal-functionalized species, *e.g.*, **{ZnV12}** (DOI: 10.1002/chem.201403592). This is now discussed in the manuscript and the ^{51}V TOCSY NMR spectrum is shown in the manuscript (Figure 2c). Also, ^{51}V NMR and ^1H NMR spectroscopic titrations have been performed which show that a 1:1 adduct between Mg^{2+} and **{V12}** (*i.e.*, **{MgV12}**) is formed (Figure 3, and Supplementary Figure 12). The speciation/fragmentation observed in the ESI MS is most likely an artefact of the ionization process and does not indicate that there is a mixture of species present in solution, as confirmed by extensive ^{51}V NMR spectroscopy. This has been observed previously in related ESI MS literature reports (*e.g.*, DOI: 10.1021/ja802514q or DOI: 10.1021/jacs.6b02245). Note that the non-metal-functionalized **{V12}** species also shows similar gas-phase speciation / fragmentation under ESI MS conditions (DOI: 10.1002/chem.201403592). These points are now discussed in the manuscript and SI (Figure 2d, Supplementary Section 2.8).

Q5: In Figure 2, the concentrations of each component (V12 and Mg salt) should be given for spectra a and c. I assume that concentrations used for 51-V NMR are much higher than those used for UV-Vis? Could this influence solution equilibria? In other words, while 51-V NMR suggests the presence of MgV12 (Fig 2a) (see point above), the existence of such species is much less obvious in UV-Vis spectrum shown in Fig 2c. I could imagine that in diluted solutions the binding between V12 and Mg(II) would be less favorable? Could changes observed in the UV-Vis spectra be related to the presence of other vanadate species detected in ESI-MS?

Reply5: the concentration information has been added. Also, the equilibria between Mg^{2+} and **{V12}** were probed by additional ^{51}V NMR spectroscopy at different Mg^{2+} and **{V12}** molar ratios and concentrations (between the UV-VIS spectroscopic concentration and the ESI MS concentration). In sum, the data show that **{MgV12}** is the only species observed by ^{51}V NMR spectroscopy, even at the

low concentrations used for ESI MS analyses ($[V_{12}] = 0.05 \text{ mM}$). This is now described in the manuscript and SI (Supplementary Figure 14).

Q6: Very little information is provided on the time scale of MgV_{12} species formation and its conversion to the final Mg_2V_{12} form. The fact that both V-51 and ESI-MS could be recorded implies that this intermediate is sufficiently stable. How quickly does it convert to Mg_2V_{12} ? According to Figure 3, it should be simple exposure to the light that should convert these species to Mg_2V_{12} ? I believe such experiment was attempted in Figure S5, but it seems that irradiation for 3 or 24 h under oxygen-free condition does not produce Mg_2V_{12} . How can this be explained? And how does this relate to the proposed mechanism in Figure 3?

Reply6: The formation of the $\{MgV_{12}\}$ intermediate by reaction of $\{V_{12}\}$ with Mg^{2+} could not be resolved on the NMR and steady-state UV-vis timescales. I.e., $\{MgV_{12}\}$ formation occurs almost instantaneously: mixing of the $\{V_{12}\}$ and Mg^{2+} reaction solutions lead to an instant colour change and instant observation of the corresponding ^{51}V NMR and ESI MS signals (within the time resolution possible for of the respective measurements (<2 min after mixing)). Stopped-flow measurements to extract quantitative kinetics are planned as future studies. The reason why $\{Mg_2V_{12}\}$ is not observed in the NMR study in Figure S5 is a solvent-dependent effect: additional experiments based on the referee comments showed that when $\{Mg_2V_{12}\}$ is dissolved in DMSO, the expected two-line ^{51}V NMR spectrum characteristic for $\{Mg_2V_{12}\}$ is observed (Supplementary Figure 5). However, dissolving $\{Mg_2V_{12}\}$ in MeCN results in the formation of an equilibrium where of one Mg^{2+} is released from the $\{Mg_2V_{12}\}$ cluster, resulting in the formation of $\{MgV_{12}\}$. This is indicated by the presence of the characteristic four-line signal in ^{51}V NMR (Supplementary Figure 6). In addition, concentration-dependent 1H NMR spectroscopy as well as 1H DOSY NMR spectroscopy were used to study the dynamic equilibrium between the binding of Mg^{2+} and DMA cations. These observations are now discussed in the manuscript and SI (Supplementary Section 3.1 / 3.2)

Q7 Figure S6 shows changes in the 1H NMR spectrum of DMA upon addition of $MgCl$ to V_{12} . Why is there such a large difference seen in the DMA peak position after irradiation of 3h or 24h, when no change is to be seen in the ^{51}V NMR spectra shown in Figure S5 for such solutions?

Reply7: Additional 1H NMR spectroscopy now shows the effects of different reaction conditions on the position of the DMA signals. In addition, concentration-dependent 1H NMR spectroscopy as well as 1H DOSY NMR spectroscopy were used to study the dynamic equilibrium between the binding of Mg^{2+} and release of DMA cations. These data are now discussed in the manuscript and SI (Supplementary Section 3.1).

Q8: Related to the point above, studying this conversion is crucial to confirm the second step (i.e. the light-driven equilibria) shown in Figure 3, as I am bit missing unambiguous evidence that step b in Figure 3 really occurs as shown here. The UV/Vis spectra shown in Fig1 indicate that irradiation of $V_{12}/MgCl_2$ mixture indeed results in formation of the reduced Mg_2V_{12} , but what is the evidence that MgV_{12} is indeed seen in this spectrum as the starting material for this conversion? (btw, there seem to be something wrong with y-axis in Figures 1c and 1d).

Reply8: the presence of $\{MgV_{12}\}$ as starting material / reagent during the irradiation has been verified by ^{51}V NMR spectroscopy (and additional ^{51}V NMR TOCSY and 1H NMR DOSY), HR ESI MS spectroscopy. These data are now presented in the manuscript (Figure 2) and SI (Supplementary Figures 9-12). Also see Reply4 and Reply6 for further details on the $\{MgV_{12}\}$ identification.

Q9: The idea of using excess of DMA to probe the equilibrium shown in Figure 3 is a clever one, but based on Le Chateliers principle the concentration of $Mg(II)$ should also have profound influence on these equilibria shown in Figure 3. Has the effect of $Mg(II)$ concentration on the equilibria in Fig 3 been investigated?

Reply9: We performed a ^{51}V NMR spectroscopic titration study which shows that $\{\text{MgV}_{12}\}$ formation is highly favoured and quantitative $\{\text{MgV}_{12}\}$ formation is observed already at 1:1 molar ratio of Mg^{2+} and $\{\text{V}_{12}\}$. This is discussed in the manuscript, see Figure 3.

Q10: The authors mention in the abstract that the addition of other cations which compete with Mg(II) can effectively inhibit the formation of metal-functionalized clusters. However, besides the experiments with DMA mentioned above, I can't seem to find any competition experiments with other metal cations. I think it would be highly interesting to explore this, also in order to prove whether it is really something "special" about Mg(II) that requires the light driven equilibria to form di-substituted V12 species, or is this phenomenon pertinent to other metal cations?

Reply10: based on the referee comments, we performed a comparative study to assess whether similar reactivity is observed when replacing Mg^{2+} with Ca^{2+} . In short, we also see the formation of $\{\text{CaV}_{12}\}$ (characteristic four-line-signal in the ^{51}V NMR) and the light-induced $\{\text{CaV}_{12}\}$ reduction (indicated by the characteristic IVCT transition). These data suggest that the reported light-induced cluster reactivity is not limited to Mg^{2+} but can be transferred to other systems. Systematic studies of other metal cations are planned for the near future. The findings for Ca^{2+} are discussed in the main manuscript and in Supplementary Section 3.4.

Q11: In few places where the concentration of Mg(II) was mentioned (for example in Figure 1 captions) it seems that Mg(II) was used in more than four-fold excess for the formation of Mg_2V_{12} , although two fold excess would have been sufficient. Why was the excess of Mg necessary? What was the ratio of Mg(II)/V12 in the experiments where the formation of the mono-substituted MgV_{12} was probed?

Reply11: the fourfold excess of Mg^{2+} is based on optimization of the reaction system for optimum yield of the single-crystalline $\{\text{Mg}_2\text{V}_{12}\}$ product. However, studies at different $\text{Mg}^{2+}/\{\text{V}_{12}\}$ molar ratios performed now have shown that $\{\text{MgV}_{12}\}$ is already formed quantitatively at a 1:1 molar ratio ($\text{Mg}^{2+}/\{\text{V}_{12}\}$), also see Reply 9. This is now described in the main manuscript.

Q12: The paper is missing in-depth discussion of this novel phenomenon. The section marked as "Discussion" is actually only a brief conclusion. A more thorough discussion should be given, as it is very intriguing that binding of Mg(II), which is a redox inactive metal, can result in formation of photoactive intermediate. What is a possible explanation/rationale for this? How does the formation of Mg_2V_{12} differ from the formation of other di-metal substituted V12 species mentioned in the introduction? Is this reaction the only one that requires presence of light and the absence of oxygen? Is the light driven equilibrium an oddity that is specific for this Mg(II)/V12 system or maybe a more general phenomenon that can be explored in the future?

Reply12: the discussion has been expanded to reflect the referee comments and to include the new data presented.

Q13: It is well known that metal substitution of POMs lowers HOMO/LUMO energy resulting in visible light photoactive POM species (ACS Catal 2018, 8 (11), 10809–10825). In that respect, theoretical calculations would have been useful to understand the nature of the novel phenomenon reported here.

Reply13: Theoretical DFT level computations have been performed, which qualitatively confirm the observed visible light photoactivity of $\{\text{MgV}_{12}\}$ and formation of the characteristic IVCT transition in the Vis/NIR region for $\{\text{Mg}_2\text{V}_{12}\}$. These data are now discussed in the manuscript and SI, Supplementary Section 4.

Reviewer #3 (Remarks to the Author):

This is an important and creative study that reports a new phenomenon in the synthesis of molecular metal oxide clusters. The photoactivation of the V12 molecular building block allows access to an otherwise inaccessible di-functionalised compound. The work has implications for synthetic chemists working in molecular metal oxide chemistry, and potentially even for those working in the preparation of extended metal oxides. The conclusions presented in this study are consistent with the data presented and the methodology appears sound. The experimental details are coherent and complete, and I am confident that the work could be recreated. I am happy to recommend publication of this paper once the following points have been addressed:

Q14: The charge of the {V12} precursor compound $(\text{NMe}_2\text{H}_2)_2[\text{V}_{12}\text{O}_{32}\text{Cl}]^{3-}$ introduced on page 2 is difficult to understand. I assume the 3- applies to the whole assembly, but this should then be clarified (perhaps with curly brackets to emphasise that this is a single anionic complex). Considering this, it might be worth including the V oxidation states in the formulae throughout the manuscript – this would help the reader to follow the redox chemistry at play.

Reply14: the vanadium oxidation states have been clarified.

Q15: The use of a visible light source feels like a bit of a blunt instrument here. Can the authors please add a comment on the choice of wavelengths used, and any wavelength-selective observations?

Reply15: the broadband visible light-emitting LED was used as (a) it has an emission maximum in the blue region where the vanadates show absorption features (Supplementary Figure 1), and (b) to avoid any wavelength-specific effects. However, we have now shown that irradiation with longer wavelengths (monochromatic LED, 470 nm), does not lead to cluster reduction, while irradiation at higher energies (monochromatic LED, 405 nm) results in cluster reduction and observation of the IVCT band. This is now described in the manuscript and shown in Supplementary Figure 17.

Q16: Page 5 – where the authors state the redox potential of $\text{O}_2/\text{H}_2\text{O}$ as +0.6 V – This E_0 value (note, this is the formal potential, not the $E_{1/2}$ value as quoted in the manuscript) is based on a system where O_2 is at 1 atm and H^+ and H_2O are at unit activity in the organic solvent (actual concentrations used were 1 M H^+ and 1 M H_2O). The authors data appears to have been collected from 1M $(\text{nBu}_4)\text{PF}_6$ in DMF. The value comparison quoted, therefore is meaningless and should be replaced with system specific data. The absence of protons in the sample voltammetry conditions means that it's likely that oxygen reduction occurs via the single electron radical pathway. It's still likely that this will reoxidize the reduced V12 compound, but real electrochemical data here would be a welcome addition.

Reply16: we have modified this section and have also collected the experimental redox potential for oxygen under our experimental conditions ($E_{1/2} = -1.28$ V, Supplementary Figure 20).

Q17: Have the authors considered what is providing the electron in the reduction of the cluster? This is an important point and should be discussed in the manuscript. Does changing the solvent system impact the reaction process? I would expect DMF to be a relatively effective electron donor and may therefore improve efficiency/rate of the reaction.

Reply17: Based on our electrochemical analysis, dimethyl ammonium, DMA, is the most likely electron donor. This is detailed in Reply 1.

Reviewers' Comments:

Reviewer #1:

Remarks to the Author:

The authors have satisfactorily addressed my concerns and I can now recommend this work for publication in Nature Communications without further revision.

Reviewer #2:

Remarks to the Author:

The authors have addressed all the concerns and questions raised in my previous referee report. They performed a series of additional experiments which further strengthen the central hypothesis of the paper. They also extensively revised the manuscript to include all new findings and discuss queries raised by the referees.

Reviewer #3:

Remarks to the Author:

The authors have made a good job of revising the paper. I'm happy to recommend it's accepted as is.

Reply to referees

REVIEWERS' COMMENTS

Reviewer #1 (Remarks to the Author):

The authors have satisfactorily addressed my concerns and I can now recommend this work for publication in Nature Communications without further revision.

Reviewer #2 (Remarks to the Author):

The authors have addressed all the concerns and questions raised in my previous referee report. They performed a series of additional experiments which further strengthen the central hypothesis of the paper. They also extensively revised the manuscript to include all new findings and discuss queries raised by the referees.

Reviewer #3 (Remarks to the Author):

The authors have made a good job of revising the paper. I'm happy to recommend it's accepted as is.

Author response:

No further scientific changes were made to the manuscript or SI.